# Clinical Prediction Models for Prognosis of Colorectal Liver Metastases: A Comprehensive Review of Regression-Based and Machine Learning Models

**DOI:** 10.3390/cancers16091645

**Published:** 2024-04-25

**Authors:** Stamatios Kokkinakis, Ioannis A. Ziogas, Jose D. Llaque Salazar, Dimitrios P. Moris, Georgios Tsoulfas

**Affiliations:** 1Department of General Surgery, School of Medicine, University Hospital of Heraklion, University of Crete, 71500 Heraklion, Greece; stamatioskokkinakis@gmail.com; 2Department of Surgery, University of Colorado Anschutz Medical Campus, Aurora, CO 80045, USA; ioannis.ziogas@cuanschutz.edu (I.A.Z.); jose.llaquesalazar@cuanschutz.edu (J.D.L.S.); 3Department of Surgery, Duke University Medical Center, Durham, NC 27710, USA; dimitrios.moris@duke.edu; 4Department of Transplantation Surgery, Centre for Research and Innovation in Solid Organ Transplantation, Aristotle University School of Medicine, 54124 Thessaloniki, Greece

**Keywords:** colorectal cancer, liver metastases, machine learning, prognosis, risk assessment

## Abstract

**Simple Summary:**

Interest in stratification of prognosis for patients with colorectal liver metastases is growing. Numerous clinical prediction models have been developed for this purpose in recent years, either with the aid of traditional statistical methods or by using the aid of artificial intelligence techniques. We herein provide an overview of relevant studies discussing the different types of predictors proven to be of importance and critically assess the variable model development and validation techniques as well as the performance of the reported models.

**Abstract:**

Colorectal liver metastasis (CRLM) is a disease entity that warrants special attention due to its high frequency and potential curability. Identification of “high-risk” patients is increasingly popular for risk stratification and personalization of the management pathway. Traditional regression-based methods have been used to derive prediction models for these patients, and lately, focus has shifted to artificial intelligence-based models, with employment of variable supervised and unsupervised techniques. Multiple endpoints, like overall survival (OS), disease-free survival (DFS) and development or recurrence of postoperative complications have all been used as outcomes in these studies. This review provides an extensive overview of available clinical prediction models focusing on the prognosis of CRLM and highlights the different predictor types incorporated in each model. An overview of the modelling strategies and the outcomes chosen is provided. Specific patient and treatment characteristics included in the models are discussed in detail. Model development and validation methods are presented and critically appraised, and model performance is assessed within a proposed framework.

## 1. Introduction

The prognosis of metastatic colorectal cancer is steadily increasing, and in patients with colorectal liver metastases (CRLMs), resectable disease is a potentially curable entity, with 5-year overall survival (OS) rates between 20 and 58% [1,2]. For potentially resectable disease, administration of neoadjuvant regimens or upfront surgery is still debatable due to limited high-quality evidence, while risk stratification, with quantification of disease burden and selection of “high-risk” patients, is becoming increasingly popular [3]. Apart from surgical resection, multiple treatment modalities, like intra-arterial treatments and local ablative techniques, are becoming available for CRLMs, highlighting the substantial interest in this patient population [2].

Interest in identifying patients with poor prognosis has led to the development of numerous prediction models, and in recent years, artificial intelligence (AI) has been used in model development through a variety of machine learning (ML) techniques, mostly in the form of radiomics-based models [4]. Despite the rise of AI, conventional statistical methods remain the cornerstone of model development, and focus has now shifted in the proper reporting of relevant studies with the release of the Transparent Reporting of a multivariable prediction model for Individual Prognosis Or Diagnosis (TRIPOD) guidelines [5]. Due to the substantial rise in relevant studies in the last decade, it is imperative to summarize the literature involving all clinical models developed for the prognostication of CRLM patients. Thus, the aim of this review is to provide an overview of the available prognostic models for patients with CRLMs with emphasis on predictors, model development and validation techniques, as well as model performance.

## 2. Traditional Prediction Models

Clinical prediction models that are still in use in many hepatobiliary units worldwide were developed in the previous decades [6,7,8,9,10,11,12,13,14,15,16,17,18,19]. Details regarding these models can be seen in Appendix A. The largest single-centre series was published in 1999 by Fong et al. and included 1001 liver resections for CRLM [7]. The classical predictors included in this model, namely nodal status of the primary tumour, disease-free interval from detection of the primary to discovery of the liver metastases < 12 months, number of tumours greater than one, preoperative carcinoembryonic antigen (CEA) level > 200 ng/mL and size of the largest tumour > 5 cm retain their significance in a large percentage of modern CRLM studies. The largest multicentre series was published from France with 1568 patients and included age, T stage, N stage, time interval from primary tumour to metastases, size of largest metastases, number of metastases and clearance (resection margin) as predictors [6]. All studies used traditional Cox regression for model development with univariate screening of predictors and did not report any measures of discrimination or calibration. Two studies utilized bootstrapping as their internal validation technique [6,19], while three studies reported an external validation of their models [9,14,15].

## 3. Methods

An extensive literature search was performed using the Pubmed database, aiming to identify all studies developing a clinical prediction model using CRLM patients, either with regression-based or ML techniques (supervised and unsupervised). External validation studies of already developed models were excluded. Additionally, reference lists of relevant reviews were also screened for potential studies. The details of the search strategy are presented in the Appendix A. The screening process aimed to identify studies focusing on the prognosis of CRLM patients, either short or long term, meaning that studies focusing on the diagnosis of liver metastases were not included in this review. Radiological predictors and studies based on radiomics were also excluded, as these models form a unique category requiring special attention. In order to provide an overview of the performance of published models, studies were included if they at least reported measures of discrimination for their models. A summary of the characteristics of the final studies included in this review is provided in Table 1 and Appendix A.

## 4. Outcome Types

A variety of short- and long-term outcomes related to the prognosis of CRLM patients have been used in the literature. The outcomes chosen for each model can be seen in Table 1. 

### 4.1. Postoperative Complications/Mortality

Prediction of postoperative complications was the goal of three studies, all of which used the Clavien–Dindo classification for categorization of the grade of complications [45,50,52,91]. One study focused on 90-day mortality following Associating Liver Partition and Portal vein Ligation for Staged hepatectomy (ALPPS) [39]. All studies developed their models using regression-based techniques, reported moderate discriminative performances and fair or good calibration.

### 4.2. Survival

Most studies (n = 59, 83.1%) focused on patient survival, either as OS, disease-specific survival (DSS), disease-free survival (DFS), cancer-specific survival (CSS), progression-free survival (PFS) or intrahepatic progression-free survival (ihPFS) (Table 1). Regarding long-term survival, two regression-based models for 10-year OS were proposed [20,57]. Over 20 significant predictors were identified in these two studies, including patient-, disease- and treatment-related variables (Appendix A). Post-recurrence survival was studied in three models [35,36,87]. Unique predictors incorporated in these models are pattern of recurrence (liver only, lung only, extrahepatic), time from hepatectomy to recurrence and treatment of the recurrent disease [36,87]. 

### 4.3. Recurrence

Ten studies developed prognostic models for prediction of recurrence [21,26,30,34,55,62,63,72,79,85]. Regression-based methods were used in seven studies [26,34,55,63,72,79,85], and the remaining three studies used ML techniques, including the least absolute shrinkage and selection operator (LASSO) [62], gradient-boosted trees (GBT) [30] and random forest (RF) with a globally optimal decision tree (OPT) analysis [21]. The latter was employed to identify the ideal margin width that minimizes the probability of intrahepatic recurrence within 5 years, and margins between 9 and 11 mm were proposed according to the diameter of the largest CRLM, the primary tumour nodal status and the primary tumour site [21]. Four studies attempted to predict early recurrence, defined either as recurrence within 6 months using a previously established definition [34,79] or by performing an additional analysis that suggested cut-offs at 11 or 13 months [55,72].

## 5. Specific Patient and Treatment Characteristics

### 5.1. Simultaneous Resections

Six recent Asian studies focused on patients undergoing simultaneous resections of the primary and metastatic tumours [37,51,52,53,72,82]. Outcomes studied included OS [53,82], CSS [82], PFS [37,51], recurrence [37,72], serious postoperative complications [52] and presence of lymph node (LN) metastases [51]. The latter was unique as an outcome of choice since a positive primary LN status is a strong predictor of poor outcomes in CRLM patients in many studies. All models were based on regression-based techniques, and notably, two of the studies performed a decision curve analysis (DCA) [72,82]. In the study by Deng et al., clinical utility was found in only a narrow range of risk thresholds [72], while Jiang et al. demonstrated superior net benefit of their model compared to AJCC stage [82].

### 5.2. Upfront Surgery versus Neoadjuvant Chemotherapy

Prediction models studying which patients will benefit from upfront surgery (UPS) instead of neoadjuvant treatment were developed in three recent studies [73,76,77]. Famularo et al. employed survival random forest (RF) to estimate the best possible treatment (BPT) for each patient [76]. Following this step, a classification and regression tree (CART) analysis was used to develop a decision tree, showing the possibility of being assigned to UPS or to a neoadjuvant regimen according to five predictors (planned R1 vascular resection, number of intrahepatic metastases, colon tumour localization, CEA and sex) [76]. He et al. used a cohort of 814 patients from the Surveillance, Epidemiology, and End Results (SEER) database to develop a logistic regression-based model using benefit from UPS (defined as survival >15 months) as their outcome [77]. The authors presented a nomogram in which a lower N stage, lower histological grade, negative CEA, chemotherapy following primary resection and absence of lung metastases were associated with a higher possibility of benefit from UPS [77]. Conversely, a study with 783 UPS patients from the US developed a nomogram for prediction of treatment failure following primary resection, defined as recurrence or death within 12 months [73]. Predictors included in the final model were primary location, interval from primary to CRLM, LN positive primary, T stage and number and size of CRLMs. Notably, continuous predictors were not dichotomized in this study before entering the nomogram [73].

### 5.3. Systemic Therapies

In the debatable field of systemic therapies for CRLMs, benefit from and response to chemotherapy were studied in recent papers. ML with an OPT analysis was employed in a multinational cohort of 1358 patients to identify which patients would benefit from adjuvant chemotherapy through a higher OS or RFS [83]. In a logistic regression model from China, disease-free interval < 12 months, tumour size, tumour number and RAS status were shown to predict major pathologic response to chemotherapy, defined as less than 50% remnant viable cells [60]. Tumour response to chemotherapy, in terms of non-progressive disease, was also an independent predictor of ihPFS in patients with unresectable disease receiving radiofrequency ablation (RFA) [66]. Two Chinese studies presented prediction models for CRLM patients receiving neoadjuvant chemotherapy followed by resection [74,89]. In this high-burden patient subgroup, an increase in tumour diameter during first-line chemotherapy was entered in the nomogram predicting DFS by Liu et al. [74], while in the model presented by Wang et al., named the tumour biology score, KRAS mutation, Fong score > 2 and poor preoperative chemotherapy response were decisive for 5-year OS [89].

### 5.4. Special Treatment Modalities (RFA, MWA, HAIP, SIRT, ALPPS)

Apart from the already mentioned study by Wu et al., in which patients undergoing RFA were enrolled [66], multiple treatment modalities, now available for CRLM patients, have been studied in recent model development papers. Qin et al. focused on patients who underwent US-guided percutaneous microwave ablation (MWA) in a selected cohort of 314 patients with a number of CRLM less than nine and size of CRLMs < 5 cm [42]. A model for ihPFS (at 1, 2 and 3 years) was developed with five predictors: maximal size of CRLM, number of CRLMs, ablative margin, primary tumour lymph node status and chemotherapy, with areas under the curve (AUCs) between 0.695 and 0.782 and fair calibration [42]. Regarding hepatic arterial infusion pump chemotherapy (HAIP chemotherapy), it has only been reported as a predictor in a Dutch model developed for 10-year OS in a cohort of patients after curative resection and/or ablation of CRLM [20]. Fendler et al. focused on patients undergoing selective internal radiation therapy (SIRT) through hepatic arterial delivery of Yttrium-90 [90Y] microspheres [80]. A nomogram incorporating four predictors, namely no prior liver surgery, CEA, transaminase toxicity ≥2.5× ULN and CRLM size ≥10 cm, demonstrated an AUC of 0.81 in the training cohort and 0.83 in an external validation cohort [80]. Lastly, in a multinational study of 486 patients receiving ALPPS as therapy for CRLM, two different models were developed using 90-day mortality after stage 2 as their target outcome [39]. The two models, using predictors available before the 1st and 2nd stage, respectively, had a moderate discrimination (AUCs: 0.70–0.72) and good calibration with the Hosmer–Lemeshow (HL) test [39].

## 6. Predictor Types

The different predictors included in the models are seen in detail in Appendix A and are depicted graphically in Figure 1.

### 6.1. Patient-Related Predictors

Few patient-related predictors are included in prediction models for patients with CRLMs. The most commonly used are patient age (14 studies), gender (5 studies), American Society of Anesthesiologists (ASA) score and comorbidities (each in 3 studies). Comorbidities were defined using the Charlson comorbidity score in two of three studies [23,83]. Body mass index (BMI), marital status and no prior liver surgery were also reported as variables (each in one study).

### 6.2. Laboratory Biomarkers

A wide variety of laboratory biomarkers have been reported as predictors. The most widely used were tumour markers, namely CEA and carbohydrate antigen 19-9 (CA19-9). CEA was included in 29 (40.8%) models, while CA19-9 was reported in 8 studies. Inflammatory markers, like C-reactive protein (CRP) and neutrophil-to-lymphocyte ratio (NLR), were used in 7% of the models, and nutritional predictors, like albumin or the Prognostic Nutrition Index (PNI), defined as serum albumin (g/L) + 5 × total lymphocyte count (10^9^/L), were also used in 7% of the models. Abnormal liver function tests (LFTs, two studies), bilirubin (two studies) and gamma-glutamyl transferase/alkaline phosphatase (GGT/ALP, three studies) were also documented to be related to poor outcomes.

### 6.3. Disease-Related Predictors

Predictors related to CRLMs or the primary disease were consistently reported in the majority of the prediction models. Number (n = 44, 62.0%) and size (n = 37, 52.1%) of the liver metastases were amongst the most essential predictors, along with N stage (n = 44, 62.0%) and location (n = 22, 31.0%) of the colorectal primary. Other important variables include synchronicity (interval between primary and metastatic tumour diagnosis, 14 studies), primary T stage (11 studies), concurrent extrahepatic disease (9 studies) and bilobar distribution of the metastases (7 studies).

### 6.4. Histopathological Predictors

Variables included in pathology reports were also of importance in several studies. Tumour grade of differentiation was the most widely used (eight studies, 11.3%), followed by lymphovascular invasion (LVI), neurovascular invasion (NVI) biliary invasion (three studies) and histopathological growth pattern (desmoplastic or not) or histologic type (two studies). Other reported predictors included vitality (percentage in pathology report), Ki-67, pMMR (mismatch repair proficiency) and SOFs (spatial organization features from histology), each in one model. The latter were identified using deep learning techniques through fully automated tissue classification and quantification of SOFs that were correlated with poorer outcomes [65].

### 6.5. Treatment-Related Predictors

Over 20 different treatment-specific predictors were included in the reported models. The most significant were surgery-related predictors (13 studies, 18.3%), like major resection (either defined as resection of greater than three or four liver segments), non-anatomical resection, bilateral resection, one- or two-stage hepatectomy and planned R1 vascular resection. Other important variables include resection margin in 12 studies (16.9%), neoadjuvant (10 studies) or adjuvant chemotherapy (6 studies), response to chemotherapy or tumour regression grade (6 studies) and postoperative complications (5 studies). Resection margin was modelled differently in the included studies, most commonly as positive or negative, dichotomized (with cut-offs at 1 or 5 mm) [24,71], trichotomized [42,67] or kept in a continuous scale in mm [57].

### 6.6. RAS Status and Molecular Predictors

Molecular predictors are gaining interest and are becoming part of clinical prediction models due to the increasing availability of sequencing techniques. The most important predictor for CRLM patients is the mutant rat sarcoma virus (RAS) oncogene, which is part of almost 30% of modern models (n = 21, 29.6%). Regarding other molecular biomarkers, Marfa et al. used the CART analysis to construct a proteomic signature in a series of 85 patients, which differentiates mild from severe cases (based on predicted OS) according to the four most significant protein peaks [81]. Another emerging predictor is differentially expressed genes (DEGs), or differentially expressed exosomal miRNAs, which has already been used in five studies [22,44,59,62,68]. After selection of candidate genes, LASSO regression, which applies a penalty to candidate predictors and eliminates some variables from the final model, was applied in all studies to construct the final gene panel. All five studies combined the DEGs with clinical predictors and reported merged scores [22,44,59,62,68].

## 7. Development and Validation Techniques

The characteristics of the included studies are summarized in Table 1. Data were collected prospectively in 10 studies (14.1%). The majority used regression-based techniques (53 studies, 74.6%), while 10 studies (14.1%) employed a mix of regression-based and ML methods. Seven models (9.9%) were developed with ML or deep learning techniques. Most studies (n = 54, 76.1%) performed a univariate screening of candidate predictors, and selected variables were entered into multivariate models. Continuous predictors were mostly dichotomized (52 studies, 73.2%) and were kept in their continuous form in only 17 studies (23.9%). Regarding internal and external validation techniques, the most frequent internal validation method was split sample (21 studies, 29.6%), followed by bootstrapping (13 studies, 18.3%) and cross-validation (7 studies, 9.9%). External validation was reported in 23 studies (32.4%), while 17 studies (23.9%) did not perform any form of validation. The most utilized method of handling missing data during model development was complete case analysis (34 studies, 47.9%), followed by multiple (12 studies, 16.9%) and single (3 studies, 4.2%) imputation. Twenty studies (28.2%) did not report how missing data were handled. Most papers (53 studies, 74.6%) divided their sample into risk groups. In 21 studies (29.6%), the calculation of model outputs for each patient led to categorization into risk groups based on the distribution of the outputs (quartiles/tertiles). In four studies (5.6%), CART or OPT analyses led to the creation of the different groups, while three studies (4.2%) utilized specific software, such as X-tile, to obtain cut-offs. The optimal combination of sensitivity and specificity was used as the criterion for cut-offs in three studies (4.2%), while all possible scores derived from simplified model versions were used in three studies (4.2%). In 19 studies (26.8%), the details of how risk groups were created were unclear. A variety of model presentation methods were reported. Nomograms (28 studies, 39.4%) and risk scores with points assigned to each predictor (23 studies, 32.4%) were the most frequently used, followed by equations (8 studies, 11.3%), online calculators (5 studies, 7.0%), CART (3 studies, 4.2%) and OPT analysis (2 studies, 2.8%).

## 8. Model Performance

Performance of the prediction models developed in the included studies is summarized in Table 2. The discriminative ability was assessed with the AUC in 70 studies and with the Akaike’s Information Criterion (AIC) in 1 study. AUCs (after internal or external validation) mostly ranged between 0.60 and 0.70 (46.5%) and 0.70 and 0.80 (38%). Good discrimination (AUCs > 0.80) was reported in only eight studies (11.3%). Notably, calibration was not reported in 30 (42.3%) studies. The most frequently reported calibration measure was calibration curves, which allow for a visual inspection of the agreement between predicted and observed events. Calibration curves were presented in 31 (50.7%) studies, followed by the HL goodness of fit test (10 studies, 14.1%), calibration slope (4 studies, 5.6%) and calibration intercept (3 studies, 4.2%). DCA, which provides an overview of risk thresholds that are expected to be useful in clinical practice, was reported in 12 studies (16.9%). 

## 9. Critical Appraisal of Published Models

Studies focusing on the prediction of patients with CRLM have increased in number in recent years. Traditional statistical techniques remain the pillar of model development and produce prognostic models that can be easily presented in detail and applied to new patients in the setting of an external validation study. ML models are apparently becoming popular; however, in the context of prognosis for CRLM, there is still a sparsity of studies when it comes to models using clinical predictors. In a recent review of AI-based models for CRLM, focusing both on diagnostic and on prognostic types of outcomes, the available models almost exclusively relied on radiomics and imaging-related predictors, while only two studies were based on clinical variables [4,26,90]. The present review focused on clinical, easily interpretable models that used predictors readily available for clinicians and documented in most contemporary databases. Even though imaging-based models, studies focusing on diagnostic outcomes and those that lacked reporting of performance measures were not presented in this review, the recent interest in predictive analytics and risk stratification of colorectal cancer patients led to the inclusion of a large number of papers solely targeting a variety of prognostic outcomes.

Several types of predictors were incorporated in the models, mostly related to the neoplastic disease or to the different treatment modalities chosen for each patient. The most influential predictors were associated with a high disease burden, like multiple and/or large metastatic lesions, originating from a node-positive primary located mostly in the ascending colon, combined with a high CEA level and a positive resection margin. These factors have already been identified in multiple studies as predictors of poor survival in CRLM patients [92,93,94]. The presence of a mutant RAS oncogene was also highlighted as a key predictor in this review, which confirms the findings of a meta-analysis showing the poor OS and RFS following resection of CRLM in mutant KRAS patients [95]. The recently emerged DEGs are promising new predictors, as components of multiple signalling pathways are shown to be correlated with a poor prognosis [96].

In the recently published Prediction model Risk Of Bias Assessment Tool (PROBAST), multiple details in the model development process are mentioned that can assist with the judgment of the risk of bias in each study [97]. Despite the fact that the majority of studies included in this review were published following the release of the TRIPOD and PROBAST guidance papers, several aspects assessed here would place most studies at a high risk of bias [5,97]. Such aspects include univariate screening of candidate predictors, poor handling of missing data, dichotomization of continuous predictors, inadequate reporting of model performance and poor assessment of optimism and overfitting. Regarding dichotomization, a large number of predictors included in the majority of studies, like number and size of CRLM, or laboratory predictors, like the CEA, were mostly dichotomized, and possible non-linear relationships between predictors and outcome in the development sample were not examined, leading to the loss of valuable information [97]. Assessment of optimism and overfitting was also problematic, due to the either frequent random splitting of the dataset, which is regarded as an inadequate method of quantifying optimism, or due to the complete absence of internal or external validation in many studies [97].

In this review, the focus was placed on assessment of model performance within a proposed framework [98]. The focus has now shifted from plain reporting of measures of discrimination to a thorough assessment with the aid of calibration metrics and DCA. The percentage of studies reporting calibration metrics was 57.7% in this review, which is relatively high compared to systematic reviews of prediction models, in which reporting is as low as 5.6%, especially when ML-based models are examined [99]. Regarding the discriminative performance of the models, the vast majority of studies reported moderate discrimination (AUC: 0.60–0.80) following internal or external validation. Similarly, a meta-analysis of prediction models for colorectal cancer patients presented pooled c-statistics between 0.57 to 0.74 for multiple survival outcomes [100]. Discrimination was significantly lower for external validation compared to development studies, indicating the need for better modelling and the proper assessment of overfitting [100].

Limitations of the present study include selection and analysis of a specific model type, namely those related to prognostic types of outcomes with clinical predictors. Exclusion of models based on radiological predictors and those aiming to promptly diagnose CRLMs was deliberate, since homogeneity in the included models was important for a proper overview and presentation. Studies incompletely reporting performance measures were also excluded, since the goal was to summarize available studies conforming to a proposed guidance. This may have led to the exclusion of studies utilizing proper model building and validation procedures if performance was not adequately assessed. Lastly, due to the fact that no formal systematic review was performed, no assessment of selection and publication bias was made, nor was there a formal assessment of within-study bias with the PROBAST tool.

## 10. Conclusions and Future Directions

Despite the rise of artificial intelligence and its popularity in all aspects of surgical oncology, research focusing on prognostication of CRLM patients is still dominated by models developed with conventional statistical techniques. The overview provided in this review can be utilized in future model development studies when selecting candidate predictors to be included in the model-building procedure. Predictors proven to be of relevance in multiple studies can be combined with other variables judged to be of clinical significance by physicians with experience in the management of CRLM patients. This method will assist in avoiding bias introduced with univariate predictor screening [97,101]. Another issue arising from this review is incomplete reporting, hampering the design of external validation studies and the proper quantification of pooled model performance in a future systematic review. Studies attempting to develop and/or externally validate prediction models for CRLM patients should adhere to the framework provided by TRIPOD [5]. Complying with such guidelines will improve the assessment of the generalizability and transportability of prediction models in a variety of different patient settings and populations.

## Figures and Tables

**Figure 1 cancers-16-01645-f001:**
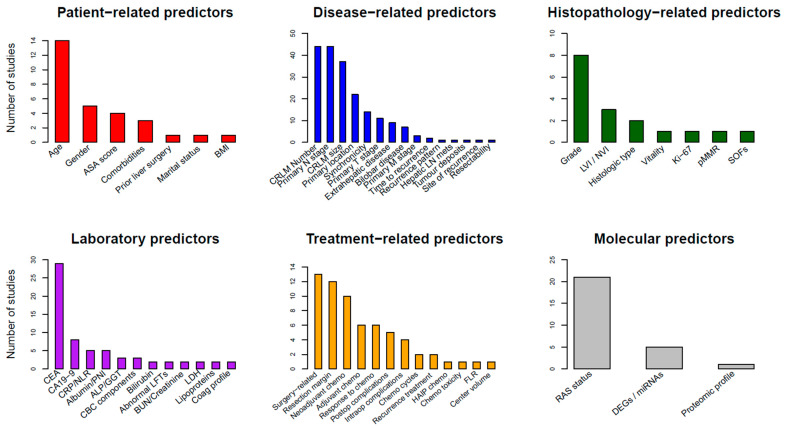
Frequency of predictors included in studies developing a clinical prediction model for prognosis of patients with colorectal liver metastases, stratified by predictor type. ASA: American Society of Anesthesiologists, BMI: body mass index, CRLM: colorectal liver metastasis, LN: lymph node, LVI: lymphovascular invasion, NVI: neurovascular invasion, pMMR: mismatch repair proficiency, SOFs: spatial organization features, CEA: carcinoembryonic antigen, CA19-9: carbohydrate antigen 19-9, PNI: prognostic nutrition index, ALP: alkaline phosphatase, GGT: gamma-glutamyl transferase, CBC: complete blood count, LFTs: liver function tests, BUN: blood urea nitrogen, LDH: lactate dehydrogenase, HAIP: hepatic arterial infusion pump, FLR: future liver remnant, RAS: rat sarcoma virus, DEGs: differentially expressed genes.

**Table 1 cancers-16-01645-t001:** Overview of model development studies for the prognosis of patients with colorectal liver metastases.

First Author (Year)	Data Collection	Model Type	Univariate Screening of Predictors	Outcome(s)	Patients (n)	Internal/External Validation	Missing Data	Risk Groups
Buisman (2022) [20]	retrospective	Cox regression	no	OS	4112	Cross-validation	Multiple imputation	yes (4 groups)
Bertsimas (2022) [21]	retrospective	RF, OPT	no	OS and intrahepatic recurrence	761	IV: Split sample/external validation cohort	Complete case analysis	no
Bao (2021) [22]	retrospective	NGS, Cox and LASSO regression	yes	OS	144	External validation (gene signature only)	No information	yes (2 groups)
Lam (2023) [23]	retrospective	Cox and LASSO regression	yes	OS and RFS	572	Split sample	Multiple imputation	no
Reijonen (2023) [24]	retrospective	Cox regression	yes	OS and DFS	816	Not performed	The final sum of their risk score points was estimated using the mean of the evaluable predictors	yes (3 groups)
Margonis (2018) [25]	retrospective	Cox regression	yes	OS	502 (development), 747 (validation)	External validation	No information	yes (3 groups)
Paredes (2020) [26]	retrospective	Mixed-effects logistic regression	no	Recurrence	703 (development), 703 (validation)	Split sample, bootstrapping	Multiple imputation	yes (3 groups)
Fruhling (2021) [27]	retrospective	Cox regression	yes	OS	1212	Cross-validation	Multiple imputation	yes (3 groups)
Taghavi (2021) [28]	retrospective	RF	no	Development of metachronous metastases	70 (development), 21 (validation)	Split sample, cross-validation	Single imputation	no
Brudvik (2019) [29]	retrospective	Cox regression	no	OS, RFS	564 (development), 608 (validation)	External validation	Complete case analysis	no
Moaven (2023) [30]	retrospective	GBT and LRB in a leave-one-out cross-validation	no	OS, recurrence	1004	Cross-validation, bootstrapping	Variables with more than 20% missing data were eliminated from the model	yes (3 groups)
Villard (2022) [31]	retrospective	Cox regression	no	OS	1013 (development), 391 (validation)	External validation	Multiple imputation	yes (4 groups)
Chen (2020) [32]	retrospective	Cox regression	no	RFS	787 (cohort 1), 162 (cohort 2)	IV: Bootstrapping/temporal validation	Complete case analysis	yes (3 groups)
Chen (2022) [33]	retrospective	Cox regression	yes	OS	1095	Not performed	Multiple imputation	no
Dai (2021) [34]	retrospective	Logistic regression	yes	Early recurrence within 6 months	150 (development), 52 (validation)	Split sample	Complete case analysis	no
Liu (2021) [35]	retrospective	Cox regression	yes	OS after recurrence	867	Bootstrapping	No information	yes (2 groups)
Liang (2021) [36]	retrospective	Cox regression	yes	Post-recurrence survival	251 (development), 125 (validation)	Split sample, bootstrapping	Complete case analysis	yes (3 groups)
Wu (2021) [37]	retrospective	Cox regression	yes	Recurrence, PFS	229 (development), 128 (validation)	Temporal validation	Complete case analysis	yes (3 groups)
Sasaki (2022) [38]	prospective	Cox regression	yes	OS	1205 (development), 1307 + 1058 (validation)	External validation	No information	yes (3 groups)
Huiskens (2019) [39]	retrospective	Logistic regression	yes	90-day mortality (after stage 2)	486	Not performed	Complete case analysis	yes (3 groups)
Bai (2022) [40]	retrospective	Cox regression	yes	OS and RFS	341 (development), 325 (validation)	External validation	Complete case analysis	yes (3 groups)
Fang (2022) [41]	retrospective	Cox regression	yes	OS	237	Not performed	Complete case analysis	yes (3 groups)
Qin (2022) [42]	prospective	Cox regression	yes	ihPFS	314	Not performed	No information	yes (3 groups)
Kawaguchi (2021) [43]	prospective	Cox regression	yes	OS	810 (development), 673 (validation)	External validation	Complete case analysis	no
Zhang (2023) [44]	retrospective	Cox and LASSO regression	yes	OS	415 (development), 207 (validation)	IV: Split sample/External validation cohort	No information	yes (2 groups)
Chen (2021) [45]	retrospective	Logistic and Cox regression	yes	Postoperative complications, PFS, OS	380	Not performed	Complete case analysis	yes (3 groups)
Jin (2022) [46]	retrospective	Cox regression	yes	CSS	881 (development), 169 (validation)	IV: Split sample/External validation cohort	Complete case analysis	yes (2 groups)
Zhai (2022) [47]	retrospective	Cox regression	yes	Liver RFS	147	Not performed	Complete case analysis	yes (3 groups)
Liu (2021) [48]	retrospective	Cox regression	yes	PFS	532 (development), 237 (validation)	External validation	No information	yes (2 groups)
Moro (2020) [49]	retrospective	CART analysis	no	OS	1123	Bootstrapping	Multiple imputation	yes (4 groups)
Chen (2021) [50]	retrospective	Logistic and Cox regression	yes	Complications, PFS, OS	169	Not performed	Complete case analysis	yes (3 groups)
Yao (2021) [51]	retrospective	Logistic and Cox regression	yes	Presence of LN metastases, PFS	241	Not performed	Complete case analysis	no
Kazi (2023) [52]	retrospective	Logistic and Cox regression	yes	Serious complications	92	Bootstrapping	No information	yes (4 groups)
Meng (2021) [53]	retrospective	Cox regression	yes	OS	174 (development), 60 (validation)	Split sample	Complete case analysis	yes (2 groups)
Imai (2016) [54]	prospective	Cox regression	yes	OS	439	Not performed	No information	yes (4 groups)
Chen (2022) [55]	retrospective	Logistic regression	yes	Early recurrence (<11 months)	144 (development), 40 (validation)	Another cohort from the same hospital	Complete case analysis	no
Cheng (2022) [56]	retrospective	Cox regression	yes	CSS	1314 (development), 560 (validation)	Split sample	Complete case analysis	yes (2 groups)
Kulik (2018) [57]	retrospective	Logistic regression	yes	OS	965	Not performed	Complete case analysis	no
Bai (2021) [58]	retrospective	Cox regression	yes	OS	490	Not performed	Complete case analysis	yes (7 and 6 groups)
Wang (2021) [59]	retrospective	Cox and LASSO regression	no	OS	113 (development), 114 (validation), 168 (external validation)	IV: Split sample/external validation cohort	Complete case analysis	yes (2 groups)
Xu (2021) [60]	retrospective	Logistic regression	yes	Major pathologic response to chemotherapy	241 (development), 241 (validation)	Split sample	Complete case analysis	yes (2 groups)
Sasaki (2018) [61]	retrospective	A priori selection of predictors and interactions	no	OS	604 (development)	External validation	No information	yes (3 groups)
Wada (2022) [62]	retrospective	Cox and LASSO regression	no	Recurrence	169 (development), 151 (validation)	External validation	No information	yes (2 groups)
Kim (2020) [63]	retrospective	Cox regression	yes	Recurrence	197 (development), 98 (validation)	Split sample	No information	yes (2 groups)
Dupre (2019) [64]	prospective	Cox regression	yes	OS	364 (development), 219 (validation)	External validation	No information	yes (2 groups)
Qi (2023) [65]	retrospective	Automated tissue classification and quantification of CRLM SOFs derived from histology images with deep learning and Cox regression	yes	OS	433 (development), 403 (validation)	External validation	Complete case analysis	yes (SOF scoring system 2 groups, SOF-CRS 3 groups)
Wu (2021) [66]	retrospective	Cox regression	yes	PFS	158	Not performed	Complete case analysis	yes (3 groups)
Dasari (2023) [67]	retrospective	Cox and LASSO regression	yes	OS	927 (development), 309 (validation)	Split sample	Complete case analysis	yes (5 groups)
Liu (2023) [68]	retrospective	Cox and LASSO regression	yes	OS	295 (development), 295 (validation)	Split sample	Complete case analysis	yes (2 groups)
Amygdalos (2023) [69]	retrospective	GBT with the Top6 selected predictors	no	OS	389 (development), 98 (validation)	Split sample	Complete case analysis	yes (2 groups)
Chen (2023) [70]	retrospective	Cox regression	yes	OS	85	Not performed	Complete case analysis	yes (3 groups)
Wu (2018) [71]	retrospective	Cox regression	yes	OS and CSS	4825 (development), 4790 (validation)	Split sample	Complete case analysis	no
Deng (2023) [72]	retrospective	Logistic regression	yes	Early recurrence (<13 months)	323 (development), 71 (validation)	External validation	Complete case analysis	no
Berardi (2023) [73]	prospective	Logistic regression	yes	Treatment failure (recurrence or death within 12 months)	535 (development), 248 (validation)	Split sample	No information	yes (2 groups)
Liu (2019) [74]	retrospective	Cox regression	yes	DFS	447 (development), 117 (validation)	External validation	No information	yes (3 groups)
Welsh (2008) [75]	prospective	Logistic regression	yes	R1 resection margin	911	Bootstrapping	Single (median) imputation	no
Famularo (2023) [76]	prospective	Survival RF to estimate the best possible treatment, then CART was used to develop a decision tree	no	OS	448	Cross-validation	Multiple imputation	yes (7 groups)
He (2023) [77]	retrospective	Logistic regression	yes	Benefit from upfront surgery (survival > 15 months)	572 (development), 242 (validation)	Split sample	Complete case analysis	no
Kattan (2008) [78]	retrospective	Cox regression	yes	DSS	1477	Bootstrapping	No information	no
Wensink (2023) [79]	retrospective	Cox regression	no	Early extrahepatic recurrence (at 6 and 12 months)	1077	Bootstrapping and internal–external cross-validation	Multiple imputation	yes (4 groups)
Fendler (2015) [80]	retrospective	Cox regression	yes	OS	100 (development), 25 (validation)	IV: Bootstrapping/external validation cohort	No information	no
Marfa (2016) [81]	prospective	CART analysis	no	OS	57 (development), 28 (validation)	Split sample	No information	yes (2 groups)
Jiang (2023) [82]	retrospective	Cox regression	yes	OSS and CSS	225 (development), 180 (validation)	External validation	Complete case analysis	no
Endo (2023) [83]	retrospective	OPT analysis	no	OS and RFS	679(development), 679 (validation)	Split sample	Multiple imputation	yes (multiple nodes)
Rees (2008) [84]	prospective	Cox regression	yes	CSS	929	Bootstrapping	Single (median) imputation	yes (5 groups)
Zakaria (2007) [85]	retrospective	Cox regression	yes	DFS, recurrence	662	Not performed	Complete case analysis	yes (3 groups)
Tan (2008) [86]	retrospective	Cox regression	yes	OS	296	Not performed	Multiple imputation	yes (3 groups)
Hill (2012) [87]	retrospective	Cox regression	yes	Survival following resection for recurrence	280	Bootstrapping	No information	yes (3 groups)
Takeda (2021) [88]	retrospective	Cox regression	yes	OS	341 (development), 309 (validation)	External validation	Complete case analysis	yes (4 groups)
Wang (2017) [89]	retrospective	Cox regression	yes	OS	300	Not performed	No information	yes (4 groups)
Spelt (2013) [90]	retrospective	ANN and Cox regression	yes	OS	241	Cross-validation	Multiple imputation	no

OS: overall survival, RF: random forest, OPT: optimal policy tree, IV: internal validation, NGS: next-generation sequencing, LASSO: least absolute shrinkage and selection operator, RFS: recurrence-free survival, DFS: disease-free survival, GBT: gradient-boosted trees, LRB: logistic regression with bootstrapping, ihPFS: intrahepatic progression-free survival, CART: classification and regression tree, LN: lymph node, CSS: cancer-specific survival, CRLM: colorectal liver metastasis, SOFs: spatial organization features, CRS: clinical risk score, ANN: artificial neural network.

**Table 2 cancers-16-01645-t002:** Performance of clinical prediction models for prognosis of patients with colorectal liver metastases.

First Author (Year)	Discrimination (AUC)	Calibration Measures	Calibration: Performance	DCA
Buisman (2022) [20]	0.73	Calibration curve	Good calibration (MSKCC model)/slight underprediction (Erasmus MC model)	NR
Bertsimas (2022) [21]	KRAS-variant: 0.76 (both training and testing)/external validation: 0.78/wild-type, training: 0.79/wild-type, testing: 0.57	NR	NR	NR
Bao (2021) [22]	Mean time-dependent: 0.75	NR	NR	NR
Lam (2023) [23]	0.65 (both for OS and RFS)	NR	NR	NR
Reijonen (2023) [24]	0.62 (OS)	NR	NR	NR
Margonis (2018) [25]	0.625	NR	NR	NR
Paredes (2020) [26]	Model without KRAS: 0.649–0.662 (validation cohort)/model with KRAS: 0.642–0.667 (validation cohort)	Calibration curve	No KRAS: good calibration/KRAS: fair	NR
Fruhling (2021) [27]	1-, 3-, 5-year OS: 0.71, 0.67, 0.67/internal validation: 0.62	Calibration curve	Excellent calibration in development cohort	NR
Taghavi (2021) [28]	Training: 0.64/validation: 0.71	NR	NR	NR
Brudvik (2019) [29]	Development, 5 -y OS: 0.69/development: 5 y RFS: 0.66	NR	NR	NR
Moaven (2023) [30]	GBT, OS: 0.77/GBT, recurrence: 0.63/LRB, OS: 0.64/LRB, recurrence: 0.57	NR	NR	NR
Villard (2022) [31]	Development: 0.74/validation: 0.69/simplified model, development: 0.74, validation: 0.66	Calibration curve, CITL, slope, HL test	CITL: 0.36, slope: 0.89 (validation), good overall fit	NR
Chen (2020) [32]	Development: 0.69 at 24 months and 0.65 at 33 months/internal validation: 0.63/cohort 2: 0.81 at 15 months	Calibration curve	Good calibration	NR
Chen (2022) [33]	1-, 3-, 5-year OS: 0.828, 0.740, 0.700 in the solitary LM group; 0.747, 0.714, 0.753 in the 2–4 LM group; 0.728, 0.741, 0.792 in the ≥ 5 LM group	Calibration curve	Fair calibration only in the 2–4 LM group	NR
Dai (2021) [34]	Training: 0.866/validation: 0.792	Calibration curve	Poor calibration in the validation cohort	Clinical utility with lift curves
Liu (2021) [35]	0.707	Calibration curve	Fair	NR
Liang (2021) [36]	Training: 0.742/validation: 0.773	Calibration curve	Fair in both training and validation cohorts	NR
Wu (2021) [37]	0.71 (both neoadjuvant and non-neoadjuvant groups)	NR	NR	NR
Sasaki (2022) [38]	Development: 0.61 (model as a continuous variable), 0.60 (model as a categorical variable)/Asian external validation cohort: 0.62 (model as a continuous variable), 0.60 (model as a categorical variable)/European external validation cohort: 0.57 (model as a continuous variable), 0.57 (model as a categorical variable)	NR	NR	NR
Huiskens (2019) [39]	Stage 1 model: 0.70/Stage 2 model: 0.72	H-L test	Stage 1 model: chi-square: 3.5, *p* = 0.63/Stage 2 model: chi-square: 7.8, *p* = 0.18	NR
Bai (2022) [40]	5-year OS, development: 0.721/5-year OS, validation: 0.665/2-year RFS, development: 0.728/2-year RFS, validation: 0.640	NR	NR	NR
Fang (2022) [41]	0.715	NR	NR	NR
Qin (2022) [42]	1-, 2-, 3-year ihPFS: 0.695, 0.764, 0.782	Calibration curve	Fair calibration	yes
Kawaguchi (2021) [43]	RAS mutant, development: 0.629/RAS mutant, validation: 0.644/wild type, development: 0.625/wild type, validation: 0.624	Calibration curve	Fair calibration (development and validation cohort)	NR
Zhang (2023) [44]	Risk score: 1, 3, 5 years, training: 0.624, 0.630, 0.662/testing: 0.610, 0.646, 0.688/validation: 0.612, 0.622, 0.652/full model: 0.783, corrected: 0.772	Calibration curve	Fair calibration	yes
Chen (2021) [45]	Complications: 0.658/PFS: 0.676/OS: 0.700	Calibration curve, HL test	Complications: fair, HL test: chi-square 3.99, *p* = 0.91/PFS: fair/OS: good	yes (for complications)
Jin (2022) [46]	Training: 0.826/validation: 0.820/external validation: 0.763	Calibration curve	Poor calibration (internal validation), fair (external validation)	yes
Zhai (2022) [47]	0.659	NR	NR	NR
Liu (2021) [48]	Development: 0.696/validation: 0.682	Calibration curve	Development: fair/validation: poor	NR
Moro (2020) [49]	AIC: wtKRAS: 1356, mtKRAS: 1356	Brier scores after bootstrapping	Brier: 0.1741 (wtKRAS), 0.1793 (mtKRAS)	NR
Chen (2021) [50]	Complications: 0.750/PFS: 0.663/OS: 0.684	Calibration curves and HL test	Complications: fair/PFS: fair/OS: fair	yes
Yao (2021) [51]	Presence of LN metastases: 0.655/PFS: 0.656	Calibration curves and HL test	Presence of LN metastases: fair/PFS: fair	NR
Kazi (2023) [52]	0.692	Calibration table	Good calibration (small group numbers)	NR
Meng (2021) [53]	1 yr OS, training: 0.788/3 yr OS, validation: 0.702/3 yr OS, training: 0.752/3 yr OS, validation: 0.848	Calibration curve	1 yr OS: fair, 3 yr OS: good (small numbers)	NR
Imai (2016) [54]	0.66	Calibration curve	3 and 5 yr OS: fair	NR
Chen (2022) [55]	Development: 0.754/validation: 0.882	Calibration curve, HL test	HL: chi-square: 1.36, *p* = 0.998, calibration curve: good calibration in development and validation cohorts	yes
Cheng (2022) [56]	Training: 0.709/validation: 0.735	Calibration curve	CSS: fair in training and validation/OS: fair in training and validation	NR
Kulik (2018) [57]	Preoperative: 0.716/preop- and perioperative: 0.761	NR	NR	NR
Bai (2021) [58]	LDH-CRS: 0.674/mCRS: 0.681	NR	NR	NR
Wang (2021) [59]	1st score, 1, 3, 5 yr OS, training: 0.84, 0.73, 0.70/1, 3, 5 yr OS, int. validation: 0.75, 0.70, 0.70/1, 3, 5 yr OS, ext. validation: 0.77, 0.78, 0.72/2nd score, 3 yr OS, training: 0.76/5 yr OS, training: 0.75/3 yr OS, validation: 0.74/5 yr OS, validation: 0.66	Calibration curve	Merged score: fair	NR
Xu (2021) [60]	Training: 0.746/validation: 0.764	Calibration curve, slope, intercept	Validation: fair, calibration slope 1.09, intercept: −0.006	NR
Sasaki (2018) [61]	0.669	NR	NR	NR
Wada (2022) [62]	Training: 0.83/validation: 0.81/mixed model: 0.85	NR	NR	NR
Kim (2020) [63]	Training: 0.824/validation: 0.898	H-L test	*p* = 0.831	NR
Dupre (2019) [64]	Preoperative: 0.619/postoperative: 0.637	NR	NR	NR
Qi (2023) [65]	SOF, 5 yr: 0.63/SOF, 8 yr: 0.74/combined, 5 yr: 0.69/combined, 8 yr: 0.79	Calibration curve	Fair calibration	NR
Wu (2021) [66]	0.705	Calibration curve	Fair calibration	NR
Dasari (2023) [67]	Development, 1, 2, 3, 5 yr: 0.756, 0.745, 0.706, 0.698/validation, 1, 2, 3, 5 yr: 0.679, 0.659, 0.678, 0.732	NR	NR	NR
Liu (2023) [68]	DEG risk score, development, 5 yr: 0.74/validation, 5 yr: 0.64/mixed model: 0.69	Calibration curve	Good calibration	yes
Amygdalos (2023) [69]	0.70	NR	NR	NR
Chen (2023) [70]	0.732	Calibration curve	Fair	NR
Wu (2018) [71]	OS, 1 and 3 yr: 0.621,0.661/CSS, 1 and 3 yr: 0.621,0.660	Calibration curve	Fair in training and validation, both for OS and CSS	NR
Deng (2023) [72]	Training: 0.720/validation: 0.740	Calibration curve, HL test	Training: fair calibration, chi-square 4.97, *p* = 0.7612/validation: poor calibration, chi: 3.89, *p* = 0.8671	yes (utility in a narrow range of thresholds)
Berardi (2023) [73]	Training: 0.68/validation: 0.60	Calibration curve	Fair	NR
Liu (2019) [74]	Development: 0.675/validation: 0.77	Calibration curve	Development: 1 yr poor, 3 yr good/validation: 1 yr poor, 3 yr poor, 5 yr poor	NR
Welsh (2008) [75]	0.781	Calibration plot, HL test	Validation: chi-square = 6.03, *p* = 0.196	NR
Famularo (2023) [76]	RF model: 0.66	NR	NR	NR
He (2023) [77]	Training: 0.801/validation: 0.739	Calibration curve, slope, intercept	Development: good calibration/validation: fair calibration, slope: 1.0, intercept 0.0	yes
Kattan (2008) [78]	Optimism-corrected: 0.612	Calibration curve	Fair	NR
Wensink (2023) [79]	Optimism-corrected, 6 m: 0.643, 12 m: 0.641	Calibration curve, slope	Fair at 6 and 12 months, optimism-corrected slope: 0.86	yes
Fendler (2015) [80]	Training 0.81/validation: 0.83	NR	NR	NR
Marfa (2016) [81]	Training: 0.903	NR	NR	NR
Jiang (2023) [82]	CSS, training, 1 and 3 yr: 0.77, 0.70/validation, 1 and 3 yr: 0.72, 0.68/OS, training, 1 and 3 yr 0.78, 0.70/validation, 1 and 3 yr: 0.74, 0.70	Calibration curve	Training: fair, validation poor	yes (superior to AJCC stage)
Endo (2023) [83]	OS-OPT, training: 0.68/testing: 0.69/RFS-OPT, training: 0.68/testing: 0.69	NR	NR	NR
Rees (2008) [84]	Preoperative: 0.781/postoperative: 0.805	H-L test	Preoperative: chi-square: 8.125; *p* = 0.087/postoperative: chi-square: 7.453, *p* = 0.114	NR
Zakaria (2007) [85]	DSS: 0.61/recurrence: 0.58	NR	NR	NR
Tan (2008) [86]	0.59	NR	NR	NR
Hill (2012) [87]	Apparent: 0.69/optimism-corrected: 0.67	NR	NR	NR
Takeda (2021) [88]	Development: 0.65	NR	NR	NR
Wang (2017) [89]	0.642	NR	NR	NR
Spelt (2013) [90]	ANN: 0.72/Cox model: 0.66	NR	NR	NR

AUC: area under the curve, DCA: decision curve analysis, MSKCC: Memorial Sloan Kettering Cancer Centre, KRAS: Kirsten rat sarcoma virus, NR: not reported, OS: overall survival, RFS: recurrence-free survival, GBT: gradient-boosted trees, LRB: logistic regression with bootstrapping, CITL: calibration-in-the-large, HL: Hosmer–Lemeshow, LM: liver metastases, ihPFS: intrahepatic progression-free survival, PFS: progression-free survival, AIC: Akaike information criterion, LN: lymph node, CSS: cancer-specific survival, LDH: lactate dehydrogenase, mCRS: modified clinical risk score, SOFs: spatial organization features, DEGs: differentially expressed genes, RF: random forest, AJCC: American Joint Committee on Cancer, OPT: optimal policy tree, DSS: disease-specific survival, ANN: artificial neural network.

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
