# Peer review of "Clinical Prediction Models for Prognosis of Colorectal Liver Metastases: A Comprehensive Review of Regression-Based and Machine Learning Models"

_cancers, 2024, doi:10.3390/cancers16091645_

Round 1
Reviewer 1 Report
Comments and Suggestions for Authors
This is an interesting review to summarize all available prognostic models of CRLM patients. I would recommend this review to publish in Cancers after minor edits.
Minor comments:
1. The author mentioned the Risk groups in Table 1, however, there is no proper explanation of how Risk groups are categorised, like 2 groups or 4 groups, etc.
2. Page14, line 322, 53 should be "53 studies".
Reviewer 2 Report
Comments and Suggestions for Authors
From a biostats and clinical epidemiology point of view, here are some comments for the Authors
- your manuscript may be considered a scoping review (do you agree!?), this info should appear in the title
- you should even briefly state the main inclusion criteria for CRLM studies, since CRLM is a quite "large" definition
- have you identified any exclusion criteria for CRLM studies? if any, state them
- have you used any selection bias evaluation for the study inclusion procedure?
- has any selected study applied the Cox PH time-dependent modeling too?
- critical point, mind to compare studies with very different median follow-up after the occurence of CRLM! can you prove that this metrics was truly comparable? median follow-up should be always reported, if available
- I wish to warmly congratulate the Authors for the very interesting infos they gave us by the supplements, especially for the covariates list of other Authors! The same fo the references, very updated!
Comments on the Quality of English Languageminor
